# Impact of Chronic Diseases on Labour Force Participation among South African Women: Further Analysis of Population-Based Data

Michael Ekholuenetale [1,*] , Anthony Ike Wegbom [2] , Clement Kevin Edet [3] , Charity Ehimwenma Joshua [4] , Amadou Barrow [5,6] and Chimezie Igwegbe Nzoputam [7,8]

1   Department of Epidemiology and Medical Statistics, Faculty of Public Health, College of Medicine, University of Ibadan, Ibadan 200284, Nigeria
2   Department of Public Health Sciences, College of Medical Sciences, Rivers State University, Port Harcourt 500101, Nigeria
3   Department of Community Medicine, College of Medical Sciences, Rivers State University, Port Harcourt 500101, Nigeria
4   Department of Economics, Faculty of Social Sciences, National Open University of Nigeria, Abuja 900107, Nigeria
5   Department of Public & Environmental Health, School of Medicine & Allied Health Sciences, University of The Gambia, Kanifing 3530, The Gambia
6   Department of Epidemiology, College of Public Health and Health Professions, University of Florida, Gainesville, FL 32610, USA
7   Department of Public Health, Center of Excellence in Reproductive Health Innovation (CERHI), College of Medical Sciences, University of Benin, Benin City 300001, Nigeria
8   Department of Medical Biochemistry, School of Basic Medical Sciences, University of Benin, Benin City 300001, Nigeria
*   Correspondence: mic42006@gmail.com

**Abstract:** The impact of chronic diseases on labour force participation is not frequently examined or considered as part of cost-of-illness studies. The aim of this study was to determine the impact of chronic diseases on labour force participation among South African women. This study included 6126 women from the 2016 South African Demographic and Health Survey. Labour force participation/employment was the outcome variable. Data were analyzed in percentage and multivariable binary logistic regression. Results showed that approximately 28.7% of women participated in the labour force and about 5.0% had diabetes. The prevalence of diabetes among women who are not in the labour force was 5.5%, whereas those in the labour force reported 3.8% prevalence of diabetes. The diabetic women had 35% reduction in labour force participation when compared with non-diabetic women (aOR = 0.65; 95% CI: 0.48 to 0.89). Geographical region was associated with labour force participation. Rural women and those currently in union/living with a man had 35% (aOR = 0.65; 95% CI: 0.56 to 0.76) and 27% (aOR = 0.73; 95% CI: 0.64 to 0.85) reduction in labour force participation, respectively, when compared with their urban and single counterparts. The findings of this study revealed that diabetes was significantly associated with reduction in labour force participation among women.

**Keywords:** diabetes; NCDs; cardiovascular diseases; health economics; women's empowerment; employment

## 1. Background

Maternal health condition is a major determinant in labour force participation as chronic diseases are contributory factors of illness-related labour market outcomes [1]. It is worthy of note that chronic diseases have gradually become the leading causes of morbidity and mortality globally. They encompass a number of illnesses such as cancers, diabetes, chronic respiratory diseases, cardiovascular disease, and musculoskeletal disorders [2].

Similarly, in sub-Saharan Africa (SSA), the problem of chronic diseases and their risk factors are also on the increase, with numerous causes which may emerge in individuals, such as hypertension [3], renal failure [4], obesity, lipiduria, and diabetes [5,6]. Diabetes and its complications cause significant economic costs for people with diabetes and their families [7]. This burden can be closely linked to societal healthcare costs involved in disease management, unintended costs that come with decreased productivity attributable to patient disability and early death, time wasted by family members supporting patients when seeking medical attention, and lost earnings.

Previous investigation revealed the connection between a variety of health-related indicators and workforce outcomes. Theoretically, health can be viewed as a human capital endowment that declines over time but is able to improve due to household production [8]. This implies that healthy workers earn higher wages and are more likely to be involved in the labour supply. Individual preferences for labour supply may be influenced by the relative utility of leisure and work as well as a reduction in the total amount of time available to earn money if they are in poor health. Furthermore, health entitles a person to income from welfare benefits if they are unable to work [9,10]. However, that is not to say that external conditions at work may contribute to physical or mental ill health in some cases, but in the case of chronic diseases, this type of bidirectional causality is unexpected.

Generally, participation in the labour force has a key role in women's lives. Besides social support, work provides income, which has positive effects on wellbeing and the quality of life [11]. Evidence-based studies show significant indirect effects of diabetes on labour force outcomes [12]. Furthermore, a large proportion of people with diabetic conditions have negative labour market supply [13]. Improved health leads to increased productivity as well as an enhanced potential to earn more money, which may contribute to greater health investment [8]. Health status, in general, has the potential to influence an individual's stock of human capital. Long periods of illness are likely to reduce work capacity and ability to participate in the labour force.

The prevalence of diabetes in African women has more than doubled, rising from 4.1% in 1980 to 8.9% in 2014 [14]. The International Diabetes Federation (IDF) has called for the establishment of national diabetes programs to better deliver prevention and control solutions in light of the rapid rise in diabetes [15], in accordance with the United Nations Sustainable Development Goal (SDG) 3.4, which aims to reduce premature mortality from noncommunicable diseases (NCDs) by one-third by 2030 [16]. Women are more affected by diabetes when compared to men, and they often take on unpaid caregiver responsibilities for affected relatives in addition to managing their own diabetes/self-care [17]. Furthermore, if a diabetic woman becomes pregnant, her foetus is at a higher risk of developing the disease in adulthood, accelerating the familial risk of diabetes [18].

There is an assertion that diabetes is attributed to long-term lifestyle factors of individuals, such as consumption of alcohol, poor diet, heredity, smoking, and physical inactivity. Furthermore, diabetes management requires patient knowledge of self-management [15]. Therefore, many countries have designed disease prevention and control programs to reduce incidence and aid patients in managing the illness [19,20]. Without proper medical attention, diabetes can result in critical medical conditions and hamper a person's ability to work and result in missed workdays (absenteeism) or lower productivity (presenteeism) [21].

Previous studies indicate that diabetes has a clear impact on economic costs [22,23], work capability, job performance, macroeconomic productivity, and socioeconomic impact [21–23]. However, most cost-of-illness studies estimate costs based on productivity losses from morbidity and mortality [24]. To understand the adverse effects of diabetes on workforce participation is painstaking. The approach to investigate the relationship between diabetes and labour force participation requires rigorous analysis of possible confounders, reverse causality between diabetes and labour force participation (endogeneity), and various stages of complications. The effects of diabetes on workforce outcomes have been reported for several high-income countries [23,25,26]. There is a need to learn about

the impact of chronic diseases on labour force participation. The research question for this study is: what impact do chronic diseases have on labour force participation among South African women?

## 2. Materials and Methods

### 2.1. Data Source

The 2016 South African Demographic and Health Survey (DHS) data was analyzed. This study included 6126 women as participants. DHS is conducted worldwide and designed to provide a wide range of health data, such as data for demographics, vaccine uptake, fertility, nutrition, family planning, sexually transmitted diseases, intimate partner violence, health behaviours, mortality amongst others as part of an important resource for monitoring population health, and vital statistics indicators. The data are used globally in public health research to determine trends, prevalence, associations, and inequalities. DHS uses a multi-stage stratified cluster random sampling approach to identify households in enumeration areas. In this study, data from the women's questionnaire was used. DHS datasets are available for researchers upon request at: http://dhsprogram.com/data/available-datasets.cfm (accessed on 9 December 2022).

### 2.2. Variable Selection and Measurement

#### 2.2.1. Outcome Variable

Labour force participation was obtained from the variable "currently working". Women were asked about their current job status, to which they responded "yes" coded as 1 or "no" coded as 0. Individuals who reported "yes" were considered as participating in the labour force.

#### 2.2.2. Exposure Variable

Respondents were asked if they were diagnosed with diabetes and other chronic diseases (high blood pressure, heart attack, cancer, stroke, high blood cholesterol, chronic bronchitis, and asthma) by a doctor/nurse, to which they responded "yes" or "no".

#### 2.2.3. Covariates

- Age (years): 15–19/20–24/25–29/30–34/35–39/40–44/45–49/50+
- Region: Western Cape/Eastern Cape/Northern Cape/Free State/Kwazulu-Natal/North West/Gauteng/Mpumalanga/Limpopo
- Residential status: urban/rural
- Education: no formal education/primary/secondary/higher
- Sex of household head: male/female
- Household wealth quintiles: DHS calculated the household wealth index as a cumulative composite of household assets using principal component analysis (PCA) and placed them on a continuous relative wealth scale [27,28]. The z-scores and factor loadings of each household asset was calculated. The loadings were multiplied by the indicator values of each household asset and then added together to calculate the wealth index value. The overall standardized z-scores were grouped to the wealth quintiles poorest/poorer/middle/richer/richest.
- Marital status: single/currently in union or living with a man/formerly in union
- Respondent perception of own health: poor/average/good/excellent
- Covered by health insurance: no/yes

### 2.3. Ethical Considerations

For this research, the investigators utilized anonymized secondary datasets available in the public domain. The authors requested access and permission was granted to use the data by MEASURE DHS/ICF International. The DHS Program complies with the industry standards to protect women's privacy during the survey. ICF International guarantees that the survey follows the United States Department of Health and Human Services'

Human Subjects Protection Act. The study did not require any additional informed consent. More details about DHS data and ethical standards are available at http://goo.gl/ny8T6X (accessed on 9 December 2022).

*2.4. Statistical Analysis*

Due to the survey design involving stratification, clustering, and weighting, the stata survey module ('svy') was used in data analyses. In addition, variance inflation factor (VIF) was used to assess collinearity at a cut-off of 10 to detect significant concerns of interdependence of variables [29]. However, there was no collinearity found. Percentage was used for univariate analysis. To estimate the adjusted odds ratios, all statistically significant variables from the unadjusted logistic regression model were included in the multivariable logistic regression model using a 20% significance level. The statistical significance was determined in the adjusted model at $p < 0.05$. Stata version 14 (StataCorp., College Station, TX, USA) was used for data analyses.

## 3. Results

Table 1 shows the distribution of chronic diseases and women's characteristics. Approximately 5% of women have diabetes. In addition, high blood pressure, heart attack, cancer, stroke, high blood cholesterol, chronic bronchitis, and asthma reported 23.7%, 4.1%, 1.2%, 1.7%, 3.3%, 1.5%, and 4.0%, respectively. Women aged 50+ years accounted for 31.6% of the total sample. Women from Kwazulu-Natal, urban women, those with secondary education, female household head, those never in union, those not covered by health insurance, and those not currently working comprised 15.8%, 54.9%, 64.1%, 62.4%, 50.2%, 86.4% of the sample, respectively.

**Table 1.** Distribution of women's characteristics.

| Variable | Number of Women | Percentage (%) |
|---|---|---|
| Diabetes | | |
| No | 5817 | 95.0 |
| Yes | 309 | 5.0 |
| High blood pressure | | |
| No | 4676 | 76.3 |
| Yes | 1450 | 23.7 |
| Heart attack | | |
| No | 5873 | 95.9 |
| Yes | 253 | 4.1 |
| Cancer | | |
| No | 6054 | 98.8 |
| Yes | 72 | 1.2 |
| Stroke | | |
| No | 6020 | 98.3 |
| Yes | 106 | 1.7 |
| High blood cholesterol | | |
| No | 5924 | 96.7 |
| Yes | 202 | 3.3 |
| Chronic bronchitis | | |
| No | 6034 | 98.5 |
| Yes | 92 | 1.5 |
| Asthma | | |
| No | 5880 | 96.0 |
| Yes | 246 | 4.0 |
| Age (in years) | | |
| 15–19 | 730 | 11.9 |
| 20–24 | 686 | 11.2 |

**Table 1.** *Cont.*

| Variable | Number of Women | Percentage (%) |
|---|---|---|
| 25–29 | 712 | 11.6 |
| 30–34 | 622 | 10.2 |
| 35–39 | 524 | 8.6 |
| 40–44 | 465 | 7.6 |
| 45–49 | 454 | 7.4 |
| 50+ | 1933 | 31.6 |
| Region | | |
| Western Cape | 474 | 7.7 |
| Eastern Cape | 798 | 13.0 |
| Northern Cape | 529 | 8.6 |
| Free State | 647 | 10.6 |
| Kwazulu-Natal | 968 | 15.8 |
| North West | 581 | 9.5 |
| Gauteng | 561 | 9.2 |
| Mpumalanga | 705 | 11.5 |
| Limpopo | 863 | 14.1 |
| Residential status | | |
| Urban | 3361 | 54.9 |
| Rural | 2765 | 45.1 |
| Education | | |
| No formal education | 586 | 9.6 |
| Primary | 1048 | 17.1 |
| Secondary | 3927 | 64.1 |
| Higher | 565 | 9.2 |
| Sex of household head | | |
| Male | 2304 | 37.6 |
| Female | 3822 | 62.4 |
| Household wealth | | |
| Poorest | 1070 | 17.5 |
| Poorer | 1226 | 20.0 |
| Middle | 1340 | 21.9 |
| Richer | 1250 | 20.4 |
| Richest | 1240 | 20.2 |
| Marital status | | |
| Never in union | 3076 | 50.2 |
| Currently in union/living with a man | 2053 | 33.5 |
| Formerly in union | 997 | 16.3 |
| Respondent perception of own health | | |
| Poor | 792 | 12.9 |
| Average | 2046 | 33.4 |
| Good | 2493 | 40.7 |
| Excellent | 795 | 13.0 |
| Covered by health insurance | | |
| No | 5295 | 86.4 |
| Yes | 831 | 13.6 |
| Currently working | | |
| No | 4371 | 71.3 |
| Yes | 1755 | 28.7 |

Table 2 shows the prevalence of diabetes and selected chronic diseases by labour force participation status and across women's characteristics. Overall, the prevalence of diabetes among women who are out of and in labour force participation were 5.5% and 3.8%, respectively. The results showed higher prevalence of diabetes and selected chronic diseases among women who are out of labour force participation. In addition, the prevalence of diabetes and selected chronic diseases increased by women's advanced age. Among women who are out of labour force participation, there was a higher prevalence of diabetes and selected chronic diseases in the urban dwellers. Furthermore, the prevalence of diabetes was higher among women from the richest household wealth quintiles, those currently in union/living with a partner, those covered by health insurance, and those who had poor perception of their own health.

Table 2. Prevalence of diabetes and selected chronic diseases across women's characteristics by labour force participation status.

| Variable | Currently Working | | | | | | | | Not Currently Working | | | | | | | |
|---|---|---|---|---|---|---|---|---|---|---|---|---|---|---|---|---|
| | Diabetes (%) | Heart Attack (%) | Cancer (%) | Stroke (%) | High Blood Cholesterol (%) | High Blood Pressure (%) | Chronic Bronchitis (%) | Asthma (%) | Diabetes (%) | Heart Attack (%) | Cancer (%) | Stroke (%) | High Blood Cholesterol (%) | High Blood Pressure (%) | Chronic Bronchitis (%) | Asthma (%) |
| **Age (in years)** | | | | | | | | | | | | | | | | |
| 15–19 | 0.0 | 5.0 | 0.0 | 0.0 | 0.0 | 5.0 | 5.0 | 5.0 | 0.3 | 0.7 | 0.3 | 0.3 | 0.4 | 1.8 | 0.4 | 2.4 |
| 20–24 | 1.9 | 2.8 | 0.0 | 0.0 | 1.9 | 3.7 | 0.0 | 1.9 | 0.2 | 0.9 | 0.2 | 0.4 | 0.5 | 3.6 | 0.7 | 2.1 |
| 25–29 | 0.4 | 1.6 | 0.0 | 0.0 | 1.6 | 8.4 | 0.0 | 2.8 | 0.0 | 1.3 | 0.0 | 0.4 | 0.2 | 7.6 | 0.4 | 2.2 |
| 30–34 | 0.8 | 1.9 | 0.0 | 1.5 | 1.5 | 10.3 | 1.2 | 2.7 | 1.4 | 2.5 | 1.4 | 1.9 | 0.6 | 9.7 | 0.6 | 5.3 |
| 35–39 | 1.7 | 2.2 | 1.3 | 2.2 | 1.3 | 10.7 | 0.4 | 0.9 | 3.1 | 2.4 | 1.7 | 1.4 | 1.7 | 17.2 | 1.0 | 3.8 |
| 40–44 | 2.3 | 2.7 | 2.7 | 1.8 | 2.3 | 22.4 | 1.8 | 3.7 | 4.9 | 2.4 | 1.6 | 0.8 | 2.4 | 20.3 | 0.4 | 2.0 |
| 45–49 | 3.4 | 4.3 | 1.3 | 2.6 | 3.8 | 28.9 | 2.1 | 2.1 | 8.7 | 7.3 | 1.8 | 1.8 | 2.7 | 24.7 | 0.5 | 5.5 |
| 50+ | 10.5 | 6.1 | 1.9 | 1.9 | 7.7 | 45.3 | 2.6 | 5.6 | 12.9 | 9.2 | 2.1 | 3.7 | 7.7 | 53.4 | 3.4 | 6.9 |
| **Region** | | | | | | | | | | | | | | | | |
| Western Cape | 6.5 | 2.2 | 3.8 | 2.7 | 8.2 | 27.7 | 3.3 | 3.3 | 9.0 | 5.2 | 2.1 | 3.1 | 12.4 | 32.8 | 5.2 | 10.0 |
| Eastern Cape | 7.5 | 3.8 | 2.4 | 0.9 | 2.4 | 27.2 | 1.4 | 2.8 | 7.4 | 5.3 | 1.4 | 2.4 | 3.9 | 29.1 | 1.0 | 8.2 |
| Northern Cape | 2.8 | 2.8 | 1.4 | 1.4 | 0.7 | 25.0 | 1.4 | 1.4 | 5.5 | 6.2 | 1.3 | 1.3 | 1.6 | 30.4 | 0.5 | 5.5 |
| Free State | 2.9 | 4.6 | 0.6 | 0.6 | 5.2 | 28.3 | 0.6 | 2.9 | 6.8 | 6.1 | 1.9 | 2.1 | 3.2 | 30.8 | 1.3 | 3.2 |
| Kwazulu-Natal | 1.8 | 1.8 | 0.4 | 3.5 | 3.1 | 14.5 | 0.0 | 2.6 | 6.8 | 1.6 | 0.4 | 1.2 | 3.0 | 17.3 | 1.6 | 3.2 |
| North West | 4.4 | 3.9 | 0.0 | 1.1 | 2.8 | 26.0 | 1.7 | 4.4 | 3.5 | 5.5 | 1.0 | 2.5 | 3.3 | 31.0 | 0.8 | 4.5 |
| Gauteng | 4.6 | 3.1 | 1.0 | 1.0 | 5.6 | 18.3 | 1.0 | 5.1 | 4.1 | 2.5 | 1.4 | 0.8 | 4.1 | 20.6 | 1.7 | 1.9 |
| Mpumalanga | 2.3 | 5.6 | 0.5 | 1.9 | 0.9 | 20.0 | 0.9 | 4.7 | 4.3 | 5.7 | 1.6 | 2.0 | 1.6 | 20.6 | 1.4 | 3.5 |
| Limpopo | 1.8 | 3.2 | 0.5 | 0.5 | 2.3 | 16.3 | 2.7 | 1.4 | 3.1 | 3.6 | 0.6 | 1.4 | 0.6 | 16.4 | 1.6 | 1.7 |
| **Residential status** | | | | | | | | | | | | | | | | |
| Urban | 4.0 | 2.8 | 1.5 | 1.7 | 4.6 | 22.2 | 1.2 | 3.5 | 6.2 | 4.5 | 1.5 | 2.1 | 5.1 | 26.8 | 1.9 | 4.7 |
| Rural | 3.5 | 4.6 | 0.5 | 1.3 | 1.3 | 22.2 | 1.8 | 2.7 | 4.9 | 4.3 | 0.9 | 1.5 | 1.4 | 21.6 | 1.2 | 4.0 |
| **Education** | | | | | | | | | | | | | | | | |
| No formal education | 4.8 | 6.0 | 1.2 | 0.0 | 2.4 | 42.9 | 0.0 | 1.2 | 10.0 | 7.2 | 1.2 | 4.2 | 4.6 | 45.0 | 2.0 | 5.0 |
| Primary | 5.6 | 6.0 | 0.9 | 3.2 | 3.7 | 27.8 | 0.9 | 5.6 | 8.8 | 8.3 | 1.3 | 2.8 | 3.7 | 39.1 | 1.9 | 5.7 |
| Secondary | 3.1 | 2.6 | 1.1 | 1.7 | 2.8 | 20.9 | 1.5 | 3.0 | 3.7 | 3.0 | 1.0 | 1.2 | 2.7 | 16.8 | 1.3 | 3.9 |
| Higher | 5.1 | 3.8 | 1.6 | 0.3 | 5.7 | 17.5 | 1.9 | 2.9 | 6.4 | 2.0 | 2.8 | 1.2 | 4.8 | 17.2 | 2.4 | 4.0 |
| **Sex of household head** | | | | | | | | | | | | | | | | |
| Male | 3.5 | 3.3 | 1.7 | 1.5 | 3.9 | 21.7 | 1.5 | 3.3 | 5.7 | 4.5 | 1.5 | 2.0 | 4.1 | 22.5 | 1.8 | 4.3 |
| Female | 4.0 | 3.5 | 0.8 | 1.6 | 3.1 | 22.4 | 1.4 | 3.1 | 5.4 | 4.4 | 1.0 | 1.7 | 2.7 | 25.3 | 1.4 | 4.4 |

**Table 2.** *Cont.*

| Variable | Currently Working | | | | | | | | Not Currently Working | | | | | | | |
|---|---|---|---|---|---|---|---|---|---|---|---|---|---|---|---|---|
| | Diabetes (%) | Heart Attack (%) | Cancer (%) | Stroke (%) | High Blood Cholesterol (%) | High Blood Pressure (%) | Chronic Bronchitis (%) | Asthma (%) | Diabetes (%) | Heart Attack (%) | Cancer (%) | Stroke (%) | High Blood Cholesterol (%) | High Blood Pressure (%) | Chronic Bronchitis (%) | Asthma (%) |
| Household wealth | | | | | | | | | | | | | | | | |
| Poorest | 1.2 | 3.7 | 1.2 | 2.4 | 0.8 | 19.5 | 0.0 | 2.0 | 2.6 | 2.8 | 0.7 | 1.1 | 1.3 | 16.6 | 1.0 | 3.8 |
| Poorer | 1.4 | 1.7 | 1.0 | 1.0 | 2.0 | 20.6 | 0.3 | 1.7 | 3.2 | 5.3 | 0.7 | 2.3 | 1.6 | 26.3 | 1.0 | 5.4 |
| Middle | 4.9 | 3.4 | 0.6 | 1.7 | 2.9 | 24.4 | 1.4 | 2.3 | 6.5 | 4.4 | 0.9 | 1.8 | 2.8 | 22.9 | 1.0 | 3.3 |
| Richer | 6.0 | 5.0 | 1.5 | 1.5 | 4.5 | 23.3 | 2.5 | 4.5 | 7.5 | 5.5 | 1.7 | 2.2 | 4.0 | 26.9 | 2.1 | 4.4 |
| Richest | 4.1 | 3.0 | 1.3 | 1.3 | 5.2 | 21.9 | 1.9 | 4.3 | 8.1 | 3.9 | 2.2 | 1.6 | 7.0 | 28.8 | 2.8 | 5.0 |
| 1-17 Marital status | | | | | | | | | | | | | | | | |
| Never in union | 2.9 | 2.4 | 1.0 | 1.3 | 2.5 | 15.6 | 0.8 | 2.7 | 2.8 | 2.3 | 0.7 | 1.1 | 1.7 | 12.9 | 1.0 | 2.6 |
| Currently in union/living with a man | 3.9 | 3.5 | 1.5 | 1.6 | 4.2 | 24.9 | 1.6 | 3.5 | 7.4 | 5.9 | 1.8 | 2.1 | 4.8 | 29.7 | 2.1 | 5.1 |
| Formerly in union | 6.1 | 6.1 | 0.7 | 2.2 | 4.0 | 34.2 | 2.9 | 4.0 | 10.9 | 8.5 | 1.7 | 3.8 | 5.4 | 50.1 | 2.2 | 8.3 |
| Respondent perception of own health | | | | | | | | | | | | | | | | |
| Poor | 9.4 | 12.6 | 1.3 | 6.9 | 5.0 | 39.6 | 2.5 | 7.6 | 12.2 | 12.3 | 2.4 | 5.5 | 4.7 | 47.2 | 3.0 | 9.3 |
| Average | 6.3 | 4.7 | 1.7 | 1.5 | 5.6 | 32.0 | 2.1 | 5.0 | 8.1 | 5.4 | 1.6 | 2.0 | 4.5 | 32.1 | 2.0 | 5.0 |
| Good | 2.2 | 1.7 | 1.1 | 1.0 | 2.2 | 16.1 | 1.1 | 1.7 | 2.2 | 1.7 | 0.6 | 0.5 | 2.3 | 14.5 | 0.9 | 2.9 |
| Excellent | 0.0 | 0.4 | 0.0 | 0.0 | 1.7 | 9.2 | 0.4 | 1.3 | 1.1 | 0.7 | 0.5 | 0.9 | 1.1 | 6.3 | 0.5 | 1.3 |
| Covered by health insurance | | | | | | | | | | | | | | | | |
| No | 3.5 | 4.1 | 0.9 | 1.6 | 3.0 | 21.6 | 1.1 | 3.0 | 5.0 | 4.3 | 1.1 | 1.7 | 2.5 | 23.5 | 1.3 | 4.2 |
| Yes | 4.9 | 1.4 | 1.9 | 1.4 | 4.7 | 23.9 | 2.6 | 3.8 | 11.1 | 5.2 | 2.0 | 2.7 | 10.9 | 31.7 | 4.2 | 5.9 |
| Total estimate | 3.8 | 3.4 | 1.1 | 1.5 | 3.4 | 22.2 | 1.4 | 3.2 | 5.5 | 4.4 | 1.2 | 1.8 | 3.3 | 24.3 | 1.5 | 4.4 |

Table 3 shows the factors associated with labour force participation. The diabetic women had 35% reduction in labour force participation when compared with non-diabetic women (aOR = 0.65; 95% CI: 0.48 to 0.89). Furthermore, advance aged, education, those having good perception of their own health, and those covered by health insurance were more likely to be out of labour force participation when compared with women aged 15–19 years. Geographical region was associated with labour force participation. Rural women and those currently in union/living with a man had 35% and 27% reduction in labour force participation, respectively, when compared with their urban and single counterparts.

**Table 3.** Diabetes and covariates associated with labour force participation among women.

| Variable | Unadjusted Odds Ratio | 95% CI | Adjusted Odds Ratio | 95% CI |
|---|---|---|---|---|
| Diabetes | | | | |
| No | 1.00 | | 1.00 | |
| Yes | 0.68 | 0.57 to 0.81 | 0.65 | 0.48 to 0.89 |
| High blood pressure | | | | |
| No | 1.00 | | 1.00 | |
| Yes | 0.89 | 0.81 to 0.97 | 0.95 | 0.80 to 1.12 |
| Heart attack | | | | |
| No | 1.00 | | 1.00 | |
| Yes | 0.77 | 0.63 to 0.93 | 0.97 | 0.70 to 1.34 |
| Cancer | | | | |
| No | 1.00 | | | |
| Yes | 0.96 | 0.68 to 1.34 | | |
| Stroke | | | | |
| No | 1.00 | | | |
| Yes | 0.84 | 0.64 to 1.13 | | |
| High blood cholesterol | | | | |
| No | 1.00 | | | |
| Yes | 1.05 | 0.86 to 1.29 | | |
| Chronic bronchitis | | | | |
| No | 1.00 | | | |
| Yes | 0.93 | 0.69 to 1.26 | | |
| Asthma | | | | |
| No | 1.00 | | 1.00 | |
| Yes | 0.73 | 0.59 to 0.88 | 0.74 | 0.53 to 1.04 |
| Age (in years) | | | | |
| 15–19 | 1.00 | | 1.00 | |
| 20–24 | 6.63 | 4.81 to 9.14 | 6.62 | 4.03 to 10.86 |
| 25–29 | 19.33 | 14.21 to 26.29 | 20.17 | 12.50 to 32.54 |
| 30–34 | 25.66 | 18.85 to 34.95 | 30.42 | 18.76 to 49.33 |
| 35–39 | 28.42 | 20.18 to 38.82 | 32.94 | 20.20 to 53.72 |
| 40–44 | 31.60 | 23.09 to 43.26 | 40.16 | 24.46 to 65.94 |
| 45–49 | 38.09 | 27.81 to 52.17 | 52.94 | 32.13 to 87.22 |
| 50+ | 10.10 | 7.49 to 13.61 | 15.94 | 9.80 to 25.91 |
| Region | | | | |
| Western Cape | 1.00 | | 1.00 | |
| Eastern Cape | 0.57 | 0.49 to 0.67 | 0.91 | 0.69 to 1.21 |
| Northern Cape | 0.59 | 0.50 to 0.70 | 0.73 | 0.54 to 0.98 |
| Free State | 0.58 | 0.49 to 0.68 | 0.68 | 0.51 to 0.90 |
| Kwazulu-Natal | 0.48 | 0.41 to 0.56 | 0.72 | 0.55 to 0.95 |
| North West | 0.71 | 0.60 to 0.84 | 1.02 | 0.76 to 1.36 |
| Gauteng | 0.85 | 0.72 to 1.01 | 0.90 | 0.68 to 1.19 |
| Mpumalanga | 0.69 | 0.59 to 0.81 | 1.17 | 0.88 to 1.56 |
| Limpopo | 0.54 | 0.46 to 0.63 | 0.94 | 0.70 to 1.26 |
| Residential status | | | | |
| Urban | 1.00 | | 1.00 | |
| Rural | 0.58 | 0.54 to 0.63 | 0.65 | 0.56 to 0.76 |

**Table 3.** *Cont.*

| Variable | Unadjusted Odds Ratio | 95% CI | Adjusted Odds Ratio | 95% CI |
|---|---|---|---|---|
| Education | | | | |
| No formal education | 1.00 | | 1.00 | |
| Primary | 1.55 | 1.30 to 1.86 | 1.44 | 1.08 to 1.92 |
| Secondary | 2.44 | 2.09 to 2.86 | 2.02 | 1.54 to 2.66 |
| Higher | 7.53 | 6.25 to 9.07 | 3.87 | 2.76 to 5.42 |
| Sex of household head | | | | |
| Male | 1.00 | | | |
| Female | 0.98 | 0.91 to 1.05 | | |
| Household wealth | | | | |
| Poorest | 1.00 | | 1.00 | |
| Poorer | 1.07 | 0.94 to 1.21 | 1.00 | 0.81 to 1.24 |
| Middle | 1.18 | 1.04 to 1.33 | 1.05 | 0.85 to 1.30 |
| Richer | 1.57 | 1.39 to 1.77 | 1.17 | 0.95 to 1.45 |
| Richest | 2.01 | 1.78 to 2.27 | 1.21 | 0.97 to 1.53 |
| Marital status | | | | |
| Never in union | 1.00 | | 1.00 | |
| Currently in union/living with a man | 1.45 | 1.34 to 1.57 | 0.73 | 0.64 to 0.85 |
| Formerly in union | 1.12 | 1.05 to 1.24 | 1.00 | 0.82 to 1.22 |
| Respondent perception of own health | | | | |
| Poor | 1.00 | | 1.00 | |
| Average | 1.42 | 1.24 to 1.61 | 1.21 | 0.98 to 1.51 |
| Good | 1.95 | 1.72 to 2.22 | 1.55 | 1.24 to 1.93 |
| Excellent | 1.70 | 1.46 to 1.98 | 1.56 | 1.19 to 2.05 |
| Covered by health insurance | | | | |
| No | 1.00 | | 1.00 | |
| Yes | 3.16 | 2.86 to 3.48 | 2.07 | 1.72 to 2.50 |

CI, confidence interval.

## 4. Discussion

The evidence suggests that diabetes has a negative impact on labour force participation. This is consistent with previous research findings [30,31]. Diabetes has direct and indirect impact on socioeconomic costs in the society due in part to reductions in productivity and participation in labour force. Moreover, in a previous study, only diabetes with complications resulted in a complete labour force exit [32]. The findings from another study revealed that people with diabetes who have complications are more likely to be unemployed than people with diabetes who do not have complications [33]. According to the available evidence, people with diabetes are more likely to leave the labour force early, retire early, or receive a permanent disability pension.

The prevalence of diabetes is increasing not only in high-income countries but also in resource-constrained settings. To address the negative impact of diseases on labour force participation and economic development, several policies and programs have been designed. One of the primary goals of the current pension reforms is to maintain and possibly extend the ability to participate in workforce of older workers. This study, however, indicates that women with diabetes may be unable to maintain their employment status and, as a result, leave the labour market sooner [15]. A significant effort should be made to improve and extend the ability of diabetics to work. Additionally, particular emphasis should be placed on developing and improving the effectiveness of evidence-based prevention and management programs.

Furthermore, we found that labour force participation increased by increasing educational attainment. Other factors that were found to be significantly associated with

labour force participation include geographical region, residential status, age, marital status, women's perception of own health, and health insurance coverage. A previous study found that higher educational attainment increases a woman's likelihood of participating in the labour force [34]. Educational training is a medium where women acquire the knowledge and skill sets to be employed. It is generally assumed that the higher an individual grows in formal education, the better chance she has to gain employment. Hence, the findings of positive association between a woman's educational attainment and labour force participation corroborate with reports from previous studies.

Ultimately, the findings of this study should be improved by looking into the underlying dynamics and strengthening the evidence to practice. To begin, future cost studies on adverse effects of diabetes or its complications on workforce outcomes should account for the complete absence of unemployment or missed workdays. This will help to forestall a gross underestimation of the burden imposed by this condition on economic development. Furthermore, future studies should examine the male population and level of disease complications as well as examine reverse causality. These methods should be used for all outcomes, not just the presence or absence of labour force participation. It is possible that other factors may have been responsible for lack of labour force participation, and it is recommended that endogeneity should be examined in future studies. This will help to identify the underlying factors and processes, whether unobserved factors or through reverse causality. That could help to understand how a chronic lifestyle illness affects workforce outcomes.

*Strengths and Limitations*

This study provided evidence on the effect of diabetes on labour force participation, including absence of employment. The use of nationally representative data, which makes the findings generalizable, is the study's main strength. In this study, however, only association, not causality, can be established. Furthermore, other aspects of the disease, such as management efficiency, health literacy, and medication adherence, were not investigated. These should have been considered in the analysis in order to gain a better understanding of the underlying factors and to allow for the individualization of concrete starting points for practical intervention. Finally, this study relied on self-report of diabetes. More objective methods of determining diabetes and an understanding of which factors and dynamics actually lead to poor labour force outcomes using various modeling strategies for comorbidities and complications should be used in future research [35,36].

**5. Conclusions**

The findings of the study show some correlation between chronic diseases and labour force participation among South African women. We found diabetes to be associated with reduction in labour force participation. We recommend concerted efforts in improving chronic diseases modelling and the understanding of how the illnesses are linked to labour force participation and outcomes. This is important not only for calculating economic costs accurately, but it may also be useful in developing evidence-based prevention and disease management programmes.

**Author Contributions:** Conceptualization, M.E., A.I.W., C.K.E., C.E.J., A.B. and C.I.N.; methodology, M.E., A.I.W., C.K.E., C.E.J., A.B. and C.I.N.; software, M.E.; validation, M.E., C.K.E., C.E.J., A.B. and C.I.N.; formal analysis, M.E. and C.E.J.; investigation, M.E., A.I.W., C.K.E., C.E.J., A.B. and C.I.N.; resources, M.E.; data curation, M.E.; writing—original draft preparation, M.E., A.I.W., C.K.E., C.E.J., A.B. and C.I.N.; writing—review and editing, M.E., A.I.W., C.K.E., C.E.J., A.B. and C.I.N.; visualization, M.E. and C.E.J.; supervision, M.E.; project administration, M.E. All authors have read and agreed to the published version of the manuscript.

**Funding:** The Inner City Fund (ICF) provided technical assistance throughout the survey program with funds from the United States Agency for International Development (USAID). However, there was no funding or sponsorship for this study or the publication of this article.

**Institutional Review Board Statement:** Ethics approval was not required for this study because the authors used secondary data that was freely available in the public domain. As a result, IRB approval is not required for this study. More information about DHS data and ethical standards can be found at: http://dhsprogram.com/data/available-datasets.cfm (accessed on 9 December 2022).

**Informed Consent Statement:** The Demographic and Health Survey is an open-source dataset that has been de-identified. As a result, the consent for publication requirement is not applicable.

**Data Availability Statement:** Data for this study were obtained from the National Demographic and Health Surveys (DHS) of the studied African countries, which can be found at http://dhsprogram.com/data/available-datasets.cfm (accessed on 9 December 2022).

**Conflicts of Interest:** The authors declare no conflict of interest.

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
