# Peer review of "Impact of Chronic Diseases on Labour Force Participation among South African Women: Further Analysis of Population-Based Data"

_world, doi:10.3390/world4010008_

Round 1
Reviewer 1 Report
Although the relationship between productivity and chronic diseases was already known, putting numbers to that relationship is always good news as it will lead to a better understanding of the disease and should serve to guide health policies.
Author Response
Reviewer #1
Although the relationship between productivity and chronic diseases was already known, putting numbers to that relationship is always good news as it will lead to a better understanding of the disease and should serve to guide health policies.
Response: Thank you very much for the insightful comment. We appreciate your observation.
Reviewer 2 Report
The paper has soundness from the analytical point of view, but still need some additions in order to be more relevant and understandable.
The title of the paper does not correspond with the analysis: as is entitle „Reducing effects of diabetes on labor force participation among working-age group of South African women: further analysis of population based data” in the content of the research are not shown the „reducing effects”, only the impact or the direct effects and not only for the diabetes. Also, the working-age group is not well explained: are the active women or inactive ones? From the data presented (table 1) it can be see that a large number of women are not currently working. In this regard, the title could not be sustained by the analysis.
From methodological point of view:
-which is the research question?
-what are the hypotheses of the research study?
-what literature sustain the research?
Concerning the used data: even if the focus of the paper is on diabetes, still the analysis presents the other illnesses were analysed. It is important to reveal why the diabetes has impact on women labor force, if this is the focus. It is a difference between women and men diabetes illnesses concerning the labor force participation? why?
The conclusions of the study should be focused on the main research question and correlated with other studies in the field.
Author Response
The paper has soundness from the analytical point of view, but still need some additions in order to be more relevant and understandable.
Response: We have now revised the paper adequately. Thank you.
The title of the paper does not correspond with the analysis: as is entitle „Reducing effects of diabetes on labor force participation among working-age group of South African women: further analysis of population based data” in the content of the research are not shown the „reducing effects”, only the impact or the direct effects and not only for the diabetes. Also, the working-age group is not well explained: are the active women or inactive ones? From the data presented (table 1) it can be see that a large number of women are not currently working. In this regard, the title could not be sustained by the analysis.
Response: We have now revised the title in line with your suggestion. Thank you for the insightful comment.
From methodological point of view:
-which is the research question?
Response: The research question sought to know what impact chronic diseases have on labour force participation among South African women. Thank you.
-what are the hypotheses of the research study?
Response: The null hypothesis would be: “there is no impact of chronic diseases on labour force participation”. As cross-sectional study design is not targeted at hypothesis testing, we did not state this needlessly. Thank you.
-what literature sustain the research?
Response: We have cited 36 references supporting literature on the correlation between chronic diseases and labour force participation. We hope the list is adequate as required by the journal guideline. Thank you.
Concerning the used data: even if the focus of the paper is on diabetes, still the analysis presents the other illnesses were analysed. It is important to reveal why the diabetes has impact on women labor force, if this is the focus. It is a difference between women and men diabetes illnesses concerning the labor force participation? why?
Response: We have revised the focus to include all chronic diseases. The study focused on women only. Our scope of the study did not include a comparison between men and women regarding diabetes. In addition, we have discussed the significant results of the study as appropriate.
The conclusions of the study should be focused on the main research question and correlated with other studies in the field.
Response: Thank you for the insightful comment. We have revised the conclusion in line with the research question.
Reviewer 3 Report
Dear Authors,
I have some comments to improve your article:
The title is not clear.
The abstract is not very clear: “about 5.0% had 39 diabetes” “prevalence of diabetes among women who are out of and in labour force were 5.5% and 3.8% respectively”.
The conclusion should be obtained by the study, in this case, “Community-based interventions to reduce the risk factors of diabetes could help address the adverse effects of diabetes in labour force participation” is not correct, which are the adverse effects of diabetes? It have not been analyze in the study, so the conclusion is not supported by the data presented in the abstract of this study.
Keywords: Do not repeat words in the title.
Better to use the symbol % instead of percent
What is the goal/s of this study? It is not well justified in the introduction, at the end “There is a need to learn about the impact of diabetes on labour force participation in resource-constrained settings such as South Africa.” If it is the goal, which was the impact?
Table 2 is not well presented, abbreviations can be used
A space is needed here: “35%and 27%”
The meaning of the abbreviation must be indicated in table 3
Best Regards
Author Response
Dear Authors,
I have some comments to improve your article:
The title is not clear.
Response: We have now revised the title as recommended. Thank you.
The abstract is not very clear: “about 5.0% had 39 diabetes” “prevalence of diabetes among women who are out of and in labour force were 5.5% and 3.8% respectively”.
The conclusion should be obtained by the study, in this case, “Community-based interventions to reduce the risk factors of diabetes could help address the adverse effects of diabetes in labour force participation” is not correct, which are the adverse effects of diabetes? It have not been analyze in the study, so the conclusion is not supported by the data presented in the abstract of this study.
Response: What we stated in the abstract was: “about 5.0% had diabetes” and unsure where “39” appeared. We have now revised the abstract as recommended. Thank you very much.
Keywords: Do not repeat words in the title.
Response: We have now revised the words as recommended. Thank you
Better to use the symbol % instead of percent
Response: This has been corrected as the percent symbol “%” has been used as recommended. Thank you
What is the goal/s of this study? It is not well justified in the introduction, at the end “There is a need to learn about the impact of diabetes on labour force participation in resource-constrained settings such as South Africa.” If it is the goal, which was the impact?
Response: We have now revised the goal of the study to determine the impact of chronic diseases on labour force participation. The impact is that diabetes was significantly associated with reduction in labour force participation among women. Thank you
Table 2 is not well presented, abbreviations can be used
Response: Table 2 was presented to show the comparative prevalence of chronic diseases between the working and non-working class women. We are unsure what exactly to abbreviate, if accepted, we hope to work with the journal’s production team to format and show the results in landscape. Thank you.
A space is needed here: “35%and 27%”
Response: We have revised as recommended. Thank you.
The meaning of the abbreviation must be indicated in table 3
Response: We have included in the foot note of table 3, the full meaning of the abbreviation – “CI, confidence interval” Thank you very much for the insightful comment.
Round 2
Reviewer 2 Report
The authors make the mandatory changes and now the paper can be publish if the other conditions of the journal are accomplished.